# RaspberrySet: Dataset of Annotated Raspberry Images for Object Detection

**Sarmīte Strautiņa** [1], **Ieva Kalniņa** [1], **Edīte Kaufmane** [1], **Kaspars Sudars** [2], **Ivars Namatēvs** [2], **Arturs Nikulins** [2] and **Edgars Edelmers** [2,*]

[1]  Institute of Horticulture, Graudu iela 1, LV-3701 Ceriņi, Latvia; sarmite.strautina@lbtu.lv (S.S.); ieva.kalnina@lbtu.lv (I.K.); edite.kaufmane@lbtu.lv (E.K.)

[2]  Institute of Electronics and Computer Science, Dzērbenes iela 14, LV-1006 Riga, Latvia; sudars@edi.lv (K.S.); ivars.namatevs@edi.lv (I.N.); arturs.nikulins@edi.lv (A.N.)

*  Correspondence: edgars.edelmers@edi.lv

**Abstract:** The RaspberrySet dataset is a valuable resource for those working in the field of agriculture, particularly in the selection and breeding of ecologically adaptable berry cultivars. This is because long-term changes in temperature and weather patterns have made it increasingly important for crops to be able to adapt to their environment. To assess the suitability of different cultivars or to make yield predictions, it is necessary to describe and evaluate berries' characteristics at various growth stages. This process is typically carried out visually, but it can be time-consuming and labor-intensive, requiring significant expert knowledge. The RaspberrySet dataset was created to assist with this process, and it includes images of raspberry berries at five different stages of development. These stages are flower buds, flowers, unripe berries, and ripe berries. All these stages of raspberry images classified buds, damaged buds, flowers, unripe berries, and ripe berries and were annotated using ground truth ROI and presented in YOLO format. The dataset includes 2039 high-resolution RGB images, with a total of 46,659 annotations provided by experts using *Label Studio* software (1.7.1). The images were taken in various weather conditions, at different times of the day, and from different angles, and they include fully visible buds, flowers, berries, and partially obscured buds. This dataset is intended to improve the efficiency of berry breeding and yield estimation and to identify the raspberry phenotype more accurately. It may also be useful for breeding other fruit crops, as it allows for the reliable detection and phenotyping of yield components at different stages of development. By providing a homogenized dataset of images taken on-site at the Institute of Horticulture in Dobele, Latvia, the RaspberrySet dataset offers a valuable resource for those working in horticulture.

**Dataset:** https://doi.org/10.5281/zenodo.7014728

**Dataset License:** CC BY 4.0.

**Keywords:** computer vision; precision horticulture; *rubus idaeus*; berry detection

## 1. Summary

Raspberry breeding at the Institute of Horticulture, Dobele, Latvia (LatHort), GPS location: N: 56°36′39″ E: 23°17′50″ has been carried out since 1980. The main objectives of raspberry breeding are to achieve the ecological plasticity of plants, high-yield and fruit quality, and resistance to diseases and pests. The structure of the raspberry cultivar in the Baltic countries has been influenced by the historical situation dominated there in the twentieth century and climatic conditions, especially the winter hardiness—commercially widely grown cultivars are mainly bred in Russia. A similar situation is observed for the genetic resources in Baltic countries, consisting of some old European and American cultivars, but mostly of Russian cultivars and hybrids. Small breeding programs are only

running in Latvia and Estonia [1]. A hybridization program provides the evaluation of about 1500 raspberry hybrids each growing season. The evaluation includes more than 30 traits. Most of them are evaluated visually, including yield compounds. Raspberries are an example of a plant with a complex set of traits influenced by the environment, i.e., meteorological conditions and genotype.

LatHort has developed rich genetic material for red raspberry, including cultivars and promising hybrids, which are intensively used in hybridization. The genotypes differ in yield compounds (number of canes, fruit laterals per cane, and the weight of fruit); winter hardiness; disease resistance; fruit quality characteristics including shape, color, biochemical composition, etc.; and fruit ripening time. Table 1 summarizes some of the most important fruit and yield component parameters of the florican raspberry cultivars and promising hybrids.

**Table 1.** Characterization of florican raspberry yield components.

| Cultivars and Hybrids | Fruit Laterals per Cane | Fruit per Fruit Lateral | The Average Weight of Fruit, g | Yield per Cane, g | Yield per Bush, g | Fruit Length, mm | Fruit Width, mm | Shape Index (Ratio Length, Width) | Account of Drupe | Fruit Glossiness (Score 1–9) | Fruit Firmness (Score 1–9) | Fruit Shape | Fruit Colour |
|---|---|---|---|---|---|---|---|---|---|---|---|---|---|
| Bozhestvennaja | 10.5 | 7.2 | 2.7 | 204.1 | 1633.0 | 23.1 | 15.6 | 1.5 | 106.2 | 2.0 | 6.0 | trapezoidal | light red |
| Glen Ample | 6.9 | 7.3 | 2.2 | 110.8 | 886.5 | 17.7 | 18.2 | 1.0 | 64.5 | 2.2 | 6.5 | broad conical | light red |
| Kapriz Bogov | 13.9 | 7.8 | 2.1 | 227.7 | 1821.5 | 20.0 | 18.7 | 1.1 | 81.1 | 4.9 | 4.0 | broad conical | red |
| Lina | 11.7 | 8.5 | 2.7 | 268.5 | 2148.1 | 17.7 | 15.8 | 1.1 | 85.5 | 3.0 | 6.0 | broad conical | light red |
| Lubetovskaja | 13.2 | 10.1 | 2.1 | 280.0 | 2239.8 | 17.4 | 15.4 | 1.1 | 71.0 | 3.7 | 5.0 | conical | dark red |
| Octavia | 8.7 | 8.8 | 2.2 | 168.4 | 1347.5 | 18.4 | 17.4 | 1.1 | 79.3 | 3.0 | 7.0 | broad conical | light red |
| Patricija | 15.6 | 8.7 | 2.3 | 312.2 | 2497.2 | 25.7 | 18.1 | 1.4 | 112.3 | 3.8 | 4.7 | trapezoidal | light red |
| Ruvi | 15.4 | 9.1 | 1.8 | 252.3 | 2018.0 | 15.8 | 14.9 | 1.1 | 77.6 | 4.0 | 5.0 | conical | light red |
| Shahrizada | 9.7 | 6 | 2.3 | 133.9 | 1070.9 | 17.7 | 15.3 | 1.2 | 86.5 | 4.2 | 6.3 | conical | dark red |
| Sulamifa | 18.6 | 7.8 | 1.3 | 188.6 | 1508.8 | 21.4 | 17.3 | 1.2 | 75.6 | 2.1 | 3.7 | trapezoidal | dark red |
| S1-12-13 | 15.4 | 9.1 | 1.8 | 252.3 | 2018.0 | 11.7 | 11.8 | 1.0 | 74.3 | 5.6 | 6.3 | conical | dark red |
| S11-25a-4 | 15.1 | 12.4 | 2.5 | 468.1 | 3744.8 | 17.3 | 16.6 | 1.0 | 80.1 | 3.8 | 4.2 | conical | red |
| S2-6-13 | 21.5 | 11.7 | 2 | 503.1 | 4024.8 | 17.0 | 15.1 | 1.1 | 94.8 | 2.9 | 5.7 | trapezoidal | red |
| S2-6-8 | 18.2 | 14.4 | 1.8 | 471.7 | 3774.0 | 19.0 | 18.2 | 1.0 | 75.5 | 2.2 | 4.4 | conical | light red |

Table 2 summarizes some of the most important fruit and yield component parameters of the primocane raspberry cultivars and promising hybrids.

**Table 2.** Characterization of primocane raspberry yield components.

| Cultivars and Hybrids | Length of Cane, cm | Length of Fruiting Part of the Cane, cm | Fruit Laterals per Cane | The Average Weight of Fruit, g | Yield per Cane, g | Yield per Bush, g | Fruit Length, mm | Fruit Width, mm | Shape Index (Ratio Length: Width) | Account of Drupe | Fruit Glossiness (Score 1–9) | Fruit Firmness (Score 1–9) | Fruit Shape | Fruit Colour |
|---|---|---|---|---|---|---|---|---|---|---|---|---|---|---|
| Brilliantovaja | 77.6 | 42.5 | 12.0 | 2.7 | 27.6 | 220.8 | 23.8 | 21.4 | 1.1 | 78.2 | 4.9 | 5.0 | conical | red |
| Gerakl | 129.1 | 57.8 | 16.1 | 2.2 | 53.0 | 424.0 | 17.3 | 19.7 | 0.9 | 56.8 | 4.4 | 7.1 | round | dark red |
| Poemat | 135.0 | 40.7 | 13.8 | 2.7 | 81.4 | 651.2 | 17.2 | 17.4 | 1.0 | 72.7 | 4.2 | 5.9 | round | light red |
| Polana | 132.8 | 47.8 | 16.0 | 2.2 | 124.7 | 997.6 | 23.4 | 20.3 | 1.2 | 108.7 | 6.7 | 5.0 | conical | red |
| Polonez | 138.1 | 35.9 | 11.8 | 2.2 | 81.1 | 649.0 | 21.5 | 18.0 | 1.2 | 103.2 | 5.6 | 6.4 | conical | light red |
| Rubinovij Gigant | 127.6 | 52.8 | 17.5 | 2.1 | 50.7 | 405.6 | 20.3 | 22.2 | 0.9 | 67.5 | 5.4 | 6.3 | broad conical | red |
| Rubinovoje Ožerelje | 125.0 | 40.7 | 13.2 | 2.1 | 84.4 | 675.2 | 24.1 | 18.9 | 1.3 | 82.2 | 4.8 | 6.7 | conical | red |
| B6R9 | 103.0 | 43.5 | 13.7 | 3.3 | 135.6 | 1084.8 | 18.1 | 18.6 | 1.0 | 66.3 | 5.2 | 4.3 | round | dark red |
| P6R3 | 1074 | 41.2 | 14.4 | 3.9 | 183.1 | 1464.8 | 24.0 | 21.6 | 1.1 | 113.9 | 8.0 | 7.0 | conical | red |
| P6R33 | 111.8 | 45.2 | 13.9 | 2.9 | 157.9 | 1263.2 | 19.6 | 19.4 | 1.0 | 70.2 | 6.9 | 6.7 | round | red |

The process of raspberry breeding takes 15–20 years from crossing to cultivar. To select candidates for cultivars, the characteristics of several thousand seedlings must be described and evaluated, most of which is performed visually. This is a time-consuming and labor-intensive process that also requires sufficient manpower. In addition, visual scoring is relatively subjective, and results may vary among different evaluators. Therefore,

the utility of new techniques for non-invasive fruit detection and phenotyping to improve yield performance should be evaluated by adopting Machine Learning (ML) techniques, considering cost–benefit and human-centered considerations.

ML and deep learning (DL) techniques have shown very promising results in fruit classification and detection problems [2] and yield quality evaluation [3]. A neat and clean image dataset in precision agriculture supplemented with an image labeling tool is the basic requirement to build accurate and robust ML models for the real-time environment.

## 2. Data Description

The annotated raspberry *rubus idaeus* dataset is a comprehensive collection of images and annotations of the fruit, specifically designed for use in the field of deep learning (DL). The dataset includes a total of 2039 original raw images, each with a resolution of 1773 × 1773 pixels, and saved in the *.jpg* format for easy accessibility and compatibility with a variety of image processing software. To provide a thorough and accurate representation of the fruit in the images, each image is accompanied by the same number of *.txt* files in the YOLO format, which stands for "You Only Look Once." (detection results from the dataset are reflected in Figure 1) [4].

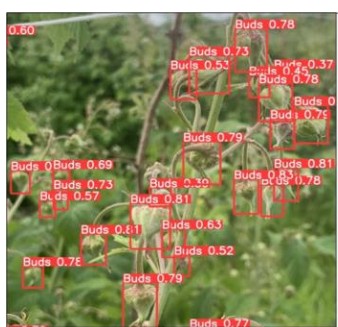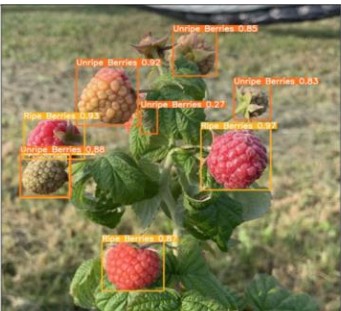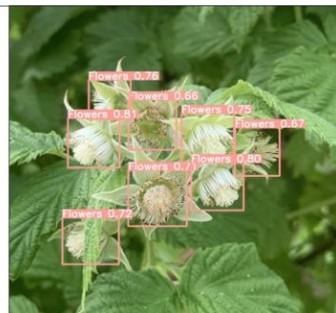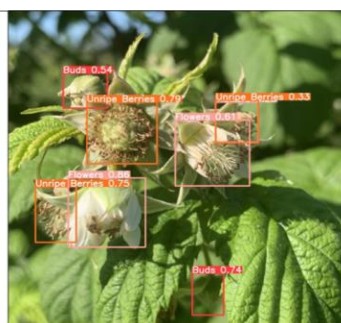

**Figure 1.** Detection results obtained with the trained detector.

The YOLO format is a popular choice in the DL community for its efficient one-level representative detection architecture, which can detect, locate, and classify the specific category of individual objects within an image. This is particularly useful in the field of plant detection, where it is necessary to quickly and accurately identify the various plant species and characteristics present in an image.

In the case of raspberry detection, the YOLO [5] format was chosen for its ability to quickly process and detect relatively small raspberry fruits. Furthermore, compared to two-level models, one-level models are generally faster at detecting and counting fruits, making them a practical choice for agricultural applications [6]. The dataset was divided into five classes: "buds", "damaged buds", "flowers", "unripe berries", and "ripe berries". The images were captured under field conditions, with images of buds, flowers, and unripe berries photographed in June 2021 and images of buds, flowers, unripe and ripe berries, and damaged buds photographed in July 2021. The images were collected from different raspberry genotypes, which can exhibit variations in bush form, yield components, and fruit location. The images were taken in an orchard at the Institute of Horticulture (LatHort) in Dobele, Latvia, by experts from LatHort who were responsible for image acquisition and manual annotation. The Institute of Electronics and Computer Science (EDI) also contributed to the creation of the dataset by providing software and hardware support. In total, out of 46,659 annotations, the raspberry dataset contains: 11,788 that were for buds, 4748 that were for flowers, 29,156 that were for unripe berries, 463 that were for ripe berries, and 504 that were for damaged buds.

## 3. Methods

All the software used is provided in Table 3.

**Table 3.** The list of software used.

| Software | Platform | Version | Information |
|---|---|---|
| *Label Studio* | 1.7.1 | https://github.com/heartexlabs/label-studio (accessed on 9 December 2022) |

### 3.1. Image Capturing

The raspberry images in the dataset were taken in an orchard at the Institute of Horticulture in Dobele, Latvia. The orchard was planted (coordinates: 56°36′23.5″ N, 23°18′09.8″ E) with 14 genotypes of raspberry, and the images were captured using an Apple XS smartphone. The images were taken at four different stages of the raspberry's phenological development: buds, flowers, unripe, and ripe fruits. The distance between the raspberry bushes and the camera was about 30 cm, capturing close-up views so that the crop elements could be seen as clearly as possible in the images. The images were taken from a variety of angles; if the lengths of the raspberry shoots were 1.0–1.4 m, then the photographing angle to the soil was 45°. If the lengths of the shoots were around 1.5–1.6 m, then the angle was 90°, but if the shoots were longer than 1.7 m, then the angle was 120°.

The images were taken under a variety of weather conditions, including sunny, cloudy, and partly cloudy. Experts from the Institute of Horticulture evaluated the images and divided them into five classes: "buds", "flowers", "unripe berries", "ripe berries", and "damaged buds". Florican raspberry buds, flowers, and unripe berry images were captured from 15 to 16 June 2021, and buds, damaged buds, flowers, and unripe and ripe berries were captured on 2 July 2021. Primocane raspberry buds, damaged buds, flowers, and unripe and ripe berries were captured on 6 August 2021 (Table 4). Temperature is one of the factors that influence yield, but when plants are grown under uncontrolled conditions (in the open field), it varies from year to year and thus affects the yield elements. For example, low temperatures during flowering can affect berry formation as the flowers are less likely to pollinate. In spring, high temperatures and insufficient moisture supply can intensify winter damage, resulting in bud dieback or the death of corroding shoots, which affects the overall view of yield elements. This may be less important for the identification of the objects themselves, but it certainly has an impact on yield and berry size. From a biological point of view, it is important that the plant characteristics are obtained under certain environmental conditions, but changing conditions, in this case, temperature, will change the yield elements and the overall characteristics. This would therefore also be relevant for yield forecasting. This could be particularly important when analyzing 3D images and comparing them with measured data.

**Table 4.** The weather conditions under which all the images were gathered.

| Date | Classes | No. of Images | Time | Air Temperature, °C | Humidity, % | PPFD, μmol/m²/s | Soil Temperature, °C | Soil Moisture Content, % |
|---|---|---|---|---|---|---|---|---|
| 15 June 2021 | "Buds", "Flowers", "Unripe Berries" | Range 1 (3516–4076)—558 images | 11:13–11:55 | 21.8 | 56.7 | 1387.8 | 18.7 | 18.5 |
| 6 June 2021 | "Buds", "Flowers", "Unripe Berries" | Range 2 (4132–4456)—324 images | 9:19–9:59 | 19.7 | 49.1 | 1472.2 | 17.5 | 15.3 |
| 2 July 2021 | "Buds", "Flowers", "Unripe Berries", "Ripe Berries", "Damaged Buds" | Range 3 (5095–5803)—678 images | 8:48–10:18 | 26.9 | 54.8 | 1430.3 | 23.6 | 9.8 |
| 6 August 2021 | "Buds", "Flowers", "Unripe Berries", "Ripe Berries", "Damaged Buds" | Range 4 (6843–7390)—512 images | 8:55–9:33 | 19.0 | 57.0 | 854.0 | 18.6 | 9.9 |

*3.2. Image Annotation*

The dataset uploaded to the Institute of Electronics and Computer Science (EDI) consists of raw images of red raspberry fruit, each saved in the .jpg format. The dataset is divided into five classes: "buds", "flowers", "unripe berries", "ripe berries", and "damaged buds". The dataset includes .txt files in the YOLO format, which provide annotations for the locations of the raspberry fruit in the images using bounding boxes. These annotations were created using the *Label Studio* software and may overlap to cover the entire berry. The YOLO format stores the annotations in .txt files: 0—buds, 1—flowers, 2—unripe berries, 3—ripe berries, and 4—damaged buds, and the following values indicate the x and y coordinates, as well as the height and width of the bounding box.

**Author Contributions:** Conceptualization, S.S.; methodology, S.S., I.K., K.S. and E.K.; software, A.N.; validation, K.S.; formal analysis, K.S.; investigation, I.N.; resources, S.S.; data curation, K.S.; writing—original draft preparation, E.E.; writing—review and editing, K.S., S.S., I.K. and E.K.; visualization, I.N.; supervision, E.K.; project administration, S.S.; funding acquisition, S.S. All authors have read and agreed to the published version of the manuscript.

**Funding:** This research and APC were funded by the Latvian Council of Science, grant number lzp-2020/1-0353 "Smart noninvasive phenotyping of raspberries and Japanese quinces using machine learning and hyperspectral and 3D imaging".

**Institutional Review Board Statement:** Not applicable.

**Informed Consent Statement:** Not applicable.

**Data Availability Statement:** The dataset is available on the Zenodo platform: https://doi.org/10.5281/zenodo.7014728 (accessed on 9 December 2022).

**Conflicts of Interest:** The authors declare no conflict of interest.

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
