# Peer review of "RaspberrySet: Dataset of Annotated Raspberry Images for Object Detection"

_data, 2023_

Round 1

Reviewer 1 Report

2. Provide more context around why certain variables were chosen for analysis in the methods section. For example, if analyzing the impact of temperature on plant growth, explain why the temperature was chosen as a variable and how it relates to the research question.

3. Consider including more visual aids, such as graphs or charts, to help illustrate key points in addition to tables and figures. For example, use a line graph to show changes in data over time or a pie chart to show proportions of different categories.

4. Expand on the limitations of the study by discussing potential sources of bias or confounding factors that may have impacted results. For example, if conducting a survey, acknowledge potential response bias due to self-selection bias.

5. Provide more information about how results compare with previous studies in this area in order to contextualize findings within existing literature. For example, cite specific studies that have found similar or different results and discuss how they relate to your own findings.

Author Response

Dear Reviewer!

  1. Provide more context around why certain variables were chosen for analysis in the methods section. For example, if analyzing the impact of temperature on plant growth, explain why the temperature was chosen as a variable and how it relates to the research question.

We have included additional information regarding into the manuscript.

  1. Consider including more visual aids, such as graphs or charts, to help illustrate key points in addition to tables and figures. For example, use a line graph to show changes in data over time or a pie chart to show proportions of different categories.

Due to absence of the temporal resolution inside the dataset it is, unfortunately, not possible to depict the data-over-time.

  1. Expand on the limitations of the study by discussing potential sources of bias or confounding factors that may have impacted results. For example, if conducting a survey, acknowledge potential response bias due to self-selection bias.

The dataset annotation is being performed manually and is one of the most crucial steps as it directly impacts the accuracy and validity of the artificial intellect systems trained on this data.

As this work is being presented and published in a form of Data Descriptor type of article, the more common sections for scientific papers such as Results, Discussion and Conclusions are absent.

  1. Provide more information about how results compare with previous studies in this area in order to contextualize findings within existing literature. For example, cite specific studies that have found similar or different results and discuss how they relate to your own findings.

As this work is being presented and published in a form of Data Descriptor type of article, the more common sections for scientific papers such as Results, Discussion and Conclusions are absent. Which in its case applies some restrictions on how much information we can put into the text.

Yours faithfully,

Edgars Edelmers

Institute of Electronics and Computer Science

Latvia, Riga, Dzērbenes Street 14

Reviewer 2 Report

  • Data description:
    • Source of the data is well documented
    • Since the dataset s publicly available data collection phase is somehow poor, but the reason why is understandable. Also authors went a litle bit further and explained the collection performed to build the dataset in a section entitled Image capturing.
    • The meta data provided knowledge provided is good and reflects on an interesting topic.
    • Copyright license is described properly and is open source.
  • Data quality:
    • The data set looks interesting for further studies. The authors documented well the using tables the attributes present.
  • Data access, archiving, and metadata
    • Data has a DOI. and is under CC by 4.0, and also reusable.

Author Response

Dear Reviewer!

Thank You for Your thoughtful comments.  We truly hope that our data will be a great use of AI solutions in agriculture.

Yours faithfully,

Edgars Edelmers

Institute of Electronics and Computer Science

Latvia, Riga, Dzērbenes Street 14

Round 2

Reviewer 1 Report

Thank you for submitting your revised manuscript. I found that you have addressed the comments and suggestions made during the initial review process, and the revised version of your manuscript has improved in terms of clarity and scientific content. Your new data provides valuable insights into object detection in agriculture.